# Spectral Filtering for
# General Linear Dynamical Systems

**Elad Hazan**
Princeton University & Google AI Princeton
ehazan@cs.princeton.edu

**Holden Lee**
Princeton University
holdenl@princeton.edu

**Karan Singh**
Princeton University & Google AI Princeton
karans@cs.princeton.edu

**Cyril Zhang**
Princeton University & Google AI Princeton
cyril.zhang@cs.princeton.edu

**Yi Zhang**
Princeton University & Google AI Princeton
y.zhang@cs.princeton.edu

## Abstract

We give a polynomial-time algorithm for learning latent-state linear dynamical systems without system identification, and without assumptions on the spectral radius of the system's transition matrix. The algorithm extends the recently introduced technique of spectral filtering, previously applied only to systems with a symmetric transition matrix, using a novel convex relaxation to allow for the efficient identification of phases.

## 1 Introduction

Linear dynamical systems (LDSs) are a cornerstone of signal processing and time series analysis. The problem of predicting the response signal arising from a LDS is a fundamental problem in machine learning, with a history of more than half a century.

An LDS is given by matrices $(A, B, C, D)$. Given a sequence of inputs $\{x_t\}$, the output $\{y_t\}$ of the system is governed by the linear equations

$$\begin{aligned} h_t &= Ah_{t-1} + Bx_t + \eta_t \\ y_t &= Ch_t + Dx_t + \xi_t, \end{aligned} \quad (1)$$

where $\eta_t, \xi_t$ are noise vectors, and $h_t$ is a hidden (latent) state.

Roweis and Ghahramani [RG99] show that special cases of this formulation capture a host of machine learning models, including hidden Markov models, Gaussian mixture models, principal component analysis, and linear Gaussian models. It has been observed numerous times in the literature that if there is no hidden state, or if the transition matrices are known, then the formulation is essentially convex and amenable to efficient optimization.

In this paper we are concerned with the general and more challenging case, arguably the one which is more applicable as well, in which the hidden state is not observed, and the system dynamics are unknown to the learner. In this setting, despite the vast literature on the subject from various communities, there is a lack of provably efficient methods for learning the LDS without strong generative or other assumptions.

Building on recent advances in spectral filtering, we develop a novel convex relaxation for LDSs, resulting in an efficient algorithm for the LDS prediction problem in the general setting. Our algorithm makes online predictions which are close (in terms of mean squared error) to those of the optimal LDS in hindsight.

## 1.1 Problem statement and our results

An LDS prediction problem is defined as follows. Iteratively for $t = 1, 2, ..., T$, the learner observes the input to the system $x_t \in \mathbb{R}^n$. The learner then makes a prediction $\hat{y}_t \in \mathbb{R}^m$, observes true outcome $y_t \in \mathbb{R}^m$, and suffers a loss $\ell(\hat{y}_t, y_t)$. For simplicity we consider the mean square error $\ell(\hat{y}_t, y_t) = \|\hat{y}_t - y_t\|^2$, even though our techniques can handle any Lipschitz convex loss.

The goal of the online learner is to minimize its regret, or difference in loss between its prediction, and the prediction of the best LDS in hindsight that predicts with $y_1^*, \ldots, y_T^*$:

$$\mathsf{Regret}(T) := \sum_{t=1}^{T} \|\hat{y}_t - y_t\|^2 - \sum_{t=1}^{T} \|y_t^* - y_t\|^2.$$

We emphasize that $y_t^*$ are not fixed vectors, but rather evolve according to a hidden state and equation (1) according to the best possible transition matrices, in terms of mean square error fit to the data.

Our main result is a polynomial-time algorithm that predicts $\hat{y}_t$ given all previous input and feedback $(x_{1:t}, y_{1:t-1})$, and attains a near-optimal regret bound of

$$\mathsf{Regret}(T) \leq \tilde{O}(\sqrt{T}) + K \cdot L.$$

Here, $L$ denotes the inevitable loss incurred by perturbations to the system which cannot be anticipated by the learner, which are allowed to be adversarial. This $L$ can grow with time, and is usually assumed to be proportional to a small constant, say $\varepsilon T$.

The constant in the $\tilde{O}(\cdot)$, as well as $K$, depend polynomially on the dimensionality of the system, the norms of the inputs and outputs, and certain natural quantities related to the transition matrix $A$. Additionally, the running time of our algorithm is polynomial in all natural parameters of the problem.

In comparison to previous approaches, we note:

- Our algorithm is the first sample-efficient and polynomial-time algorithm with this guarantee. In the next section, we survey local search algorithms that either only converge to local optima or require an exponential number of iterations in the worst case.
- The main feature is that the regret does not depend on the spectral radius $\rho(A)$ of the system's hidden-state transition matrix. If one allows a dependence on the condition number, then simple linear regression-based algorithms are known to obtain the same result, with time and sample complexity polynomial in $\frac{1}{1-\rho(A)}$. (See Section 6 of [HMR16].)

## 1.2 Related work

The prediction problems of time series for linear dynamical systems was defined in the seminal work of Kalman [Kal60], who introduced the Kalman filter as a recursive least-squares solution for maximum likelihood estimation (MLE) of Gaussian perturbations to the system. For more background see the classic survey [Lju98], and the extensive overview of recent literature in [HMR16].

For a linear dynamical system with no hidden state, the system is identifiable by a convex program and thus well understood (see [DMM$^+$17, AYS11], who address sample complexity issues and regret for system identification and linear-quadratic control in this setting).

Various exponential-time approaches have been proposed to learn the system in the case that the system is unknown. Regret bounds similar to ours are obtainable using the continuous multiplicative-weights algorithm (see [CBL06], as well as the EWOO algorithm in [HAK07]). These methods, mentioned briefly in [HSZ17], basically amount to discretizing the entire parameter space of LDSs, and take time exponential in the system dimensions. Stronger guarantees are obtained in [KM17], though still in exponential time.

Ghahramani and Roweis [RG99] suggest using the EM algorithm to learn the parameters of an LDS. This approach remains widely used, but is inherently non-convex and can get stuck in local minima. Recently [HMR16] show that for a restricted class of systems, gradient descent (also widely used in practice, perhaps better known in this setting as backpropagation) guarantees polynomial convergence rates and sample complexity in the batch setting. Their result applies essentially only to the SISO case, depends polynomially on the spectral gap, and requires the signal to be generated by an LDS.

In recent work, [HSZ17] show how to efficiently learn an LDS in the online prediction setting, without any generative assumptions, and without dependence on the condition number. Their new methodology, however, was restricted to LDSs with symmetric transition matrices. For the structural result, we use the same results from the spectral theory of Hankel matrices; see [BT17, Hil94, Cho83]. Obtaining provably efficient algorithms for the general case is significantly more challenging.

We make use of linear filtering, or linear regression on the past observations as well as inputs, as a subroutine for future prediction. This technique is well-established in the context of autoregressive models for time-series prediction that have been extensively studied in the learning and signal-processing literature, see e.g. [Ham94, BJR94, BD09, KM16, AHMS13, MW07].

The recent success of recurrent neural networks (RNNs) for tasks such as speech and language modeling has inspired a resurgence of interest in linear dynamical systems [HMR16, BK15].

## 2   Preliminaries

### 2.1   Setting

A *linear dynamical system* $\Theta = (A, B, C, D)$, with initial hidden state $h_0 \in \mathbb{R}^d$, specifies a map from *inputs* $x_1, \ldots, x_T \in \mathbb{R}^n$ to *outputs* (responses) $y_1, \ldots, y_T \in \mathbb{R}^m$, given by the recursive equations

$$h_t = Ah_{t-1} + Bx_t + \eta_t \tag{2}$$
$$y_t = Ch_t + Dx_t + \xi_t, \tag{3}$$

where $A, B, C, D$ are matrices of appropriate dimension, and $\eta_t, \xi_t$ are noise vectors.

We make the following assumptions to characterize the "size" of an LDS we are competing against:

1. Inputs and outputs and bounded: $\|x_t\|_2 \leq R_x, \|y_t\|_2 \leq R_y$.[1]

2. The system is Lyapunov stable, i.e., the largest singular value of $A$ is at most 1: $\rho(A) \leq 1$. Note that we do not need this parameter to be bounded away from 1.

3. $A$ is diagonalizable by a matrix with small entries: $A = \Psi \Lambda \Psi^{-1}$, with $\|\Psi\|_F \|\Psi^{-1}\|_F \leq R_\Psi$. Intuitively, this holds if the eigenvectors corresponding to larger eigenvalues aren't close to linearly dependent.

4. $B, C, D$ have bounded spectral norms: $\|B\|_2, \|C\|_2, \|D\|_2 \leq R_\Theta$.

5. Let $S = \left\{ \frac{\alpha}{|\alpha|} : \alpha \text{ is an eigenvalue of } A \right\}$ be the set of phases of all eigenvalues of $A$. There exists a monic polynomial $p(x)$ of degree $\tau$ such that $p(\omega) = 0$ for all $\omega \in S$, the $L^1$ norm of its coefficients is at most $R_1$, and the $L^\infty$ norm is at most $R_\infty$. We will explain this condition in Section 4.1.

In our regret model, the adversary chooses an LDS $(A, B, C, D)$, and has a budget $L$. The dynamical system produces outputs given by the above equations, where the noise vectors $\eta_t, \xi_t$ are chosen adversarially, subject to a budget constraint: $\sum_{t=1}^{T} \|\eta_t\|^2 + \|\xi_t\|^2 \leq L$.

Then, the online prediction setting is identical to that proposed in [HSZ17]. For each iteration $t = 1, \ldots, T$, the input $x_t$ is revealed, and the learner must predict a response $\hat{y}_t$. Then, the true $y_t$ is revealed, and the learner suffers a *least-squares* loss of $\|y_t - \hat{y}_t\|^2$. Of course, if $L$ scales with

the time horizon $T$, it is information-theoretically impossible for an online algorithm to incur a loss sublinear in $T$, even under non-adversarial (e.g. Gaussian) perturbations. Thus, our end-to-end goal is to track the LDS with loss that scales with the total magnitude of the perturbations, independently of $T$.

This formulation is fundamentally a min-max problem: given a limited budget of perturbations, an adversary tries to maximize the error of the algorithm's predictions, while the algorithm seeks to be robust against any such adversary. This corresponds to the $H^\infty$ notion of robustness in the control theory literature; see Section 15.5 of [ZDG+96].

## 2.2 Spectral filtering for time series

The *spectral filtering* technique is introduced in [HSZ17], which considers a spectral decomposition of the derivative of the impulse response function of an LDS with a symmetric transition matrix. A crucial object of consideration in spectral filtering is the set of *wave-filters* $\phi_1, \ldots, \phi_k$, which are the top $k$ eigenvectors of the deterministic Hankel matrix $Z_T \in \mathbb{R}^{T \times T}$, whose entries are given by $Z(i,j) = \frac{2}{(i+j)^3 - (i+j)}$. Bounds on the $\varepsilon$-rank of positive semidefinite Hankel matrices can be found in [BT17]. Our algorithm will "compress" the input time series using a time-domain convolution of the input time series with filters derived from these eigenvectors.

## 2.3 Notation for matrix norms

We will consider a few "mixed" $\ell_p$ matrix norms of a 4-tensor $M$, whose elements are indexed by $M(p, h, j, i)$ (the roles and bounds of these indices will be introduced later). For conciseness, whenever the norm of such a 4-tensor is taken, we establish the notation for the mixed matrix norm

$$\|M\|_{2,q} := \left[ \sum_p \left( \sum_{h,i,j} M(p,h,j,i)^2 \right)^{q/2} \right]^{1/q},$$

and the limiting case

$$\|M\|_{2,\infty} := \max_p \sqrt{\sum_{h,i,j} M(p,h,j,i)^2}.$$

These are the straightforward analogues of the matrix norms defined in [KSST12], and appear in the regularization of the online prediction algorithm.

# 3   Algorithm and main theorem

We begin by describing the algorithm in terms of a linear model $y(\hat{\Theta}_t; x_{1:t}; y_{t-1:t-\tau})$, the details of which occur in Definition 2.

---
**Algorithm 1** Phased wave-filtered regression
---
1: Input: time horizon $T$, parameters $k, W, \tau, R_{\hat{\Theta}}$, regularization weight $\eta$.
2: Compute $\{(\sigma_j, \phi_j)\}_{j=1}^k$, the top $k$ eigenpairs of $Z_T$.
3: Initialize $\hat{\Theta}_1 \in \mathcal{K}$ arbitrarily.
4: **for** $t = 1, \ldots, T$ **do**
5:    Predict $\hat{y}_t := y(\hat{\Theta}_t; x_{1:t}; y_{t-1:t-\tau})$.
6:    Observe $y_t$. Suffer loss $\|y_t - \hat{y}_t\|^2$.
7:    Solve FTRL convex program:

$$\hat{\Theta}_{t+1} \leftarrow \underset{\hat{\Theta} \in \mathcal{K}}{\arg\min} \sum_{u=0}^{t-1} \|y(\hat{\Theta}; x_{1:u}, y_{u-1:u-\tau}) - y_u\|^2 + \frac{1}{\eta} R(\hat{\Theta}).$$

8: **end for**

---

The central result in the paper is stated below.

**Theorem 1** (Main; informal). *Consider a LDS with noise (given by (2) and (3)) satisfying the assumptions in Section 2.1, where total noise is bounded by $L$. Then there is a choice of parameters such that Algorithm 1 learns a linear model $\hat{\Theta}$ whose predictions $\hat{y}_t$ satisfy*

$$\sum_{t=1}^{T} \|\hat{y}_t - y_t\|^2 \leq \tilde{O}\left(\text{poly}(R, d')\sqrt{T} + R_\infty^2 \tau^3 R_\Theta^2 R_\Psi^2 L\right) \tag{4}$$

*where $R_1, R_x, R_y, R_\Theta, R_\Psi \leq R, m, n, d \leq d'$.*

To define the algorithm, we specify a reparameterization of linear dynamical systems. To this end, we define a *pseudo-LDS*, which pairs a subspace-restricted linear model of the impulse response with an autoregressive model:

**Definition 2.** A *pseudo-LDS* $\hat{\Theta} = (M, N, \beta, P)$ is given by two 4-tensors $M, N \in \mathbb{R}^{W \times k \times n \times m}$ a vector $\beta \in \mathbb{R}^\tau$, and matrices $P_0, \ldots, P_{\tau-1} \in \mathbb{R}^{m \times n}$. Let the *prediction* made by $\hat{\Theta}$, which depends on the entire history of inputs $x_{1:t}$ and $\tau$ past outputs $y_{t-1:t-\tau}$ be given by

$$y(\hat{\Theta}; x_{1:t}, y_{t-1:t-\tau})(:) := \sum_{u=1}^{\tau} \beta_u y_{t-u} + \sum_{j=0}^{\tau-1} P_j x_{t-j}$$

$$+ \sum_{p=0}^{W-1} \sum_{i=1}^{n} \sum_{h=1}^{k} \sum_{u=\tau}^{t} \left[ \left( M(p, h, i, :) \cos\left(\frac{2\pi up}{W}\right) + N(p, h, i, :) \sin\left(\frac{2\pi up}{W}\right) \right) \sigma_h^{\frac{1}{4}} \phi_h(u) x_{t-u}(i) \right]$$

Here, $\phi_1, \ldots, \phi_k \in \mathbb{R}^T$ are the top $k$ eigenvectors, with eigenvalues $\sigma_1, \ldots, \sigma_k$, of $Z_T$. These can be computed using specialized methods [BLV98]. Some of the dimensions of these tensors are parameters to the algorithm, which we list here:

- Number of filters $k$.
- Phase discretization parameter $W$.
- Autoregressive parameter $\tau$.

Additionally, we define the following:

- Regularizer $R(M, N, \beta, P) := \|M\|_{2,q}^2 + \|N\|_{2,q}^2 + \|\beta\|_{q'}^2 + \sum_{j=1}^{\tau} \|P_j\|_F^2$, where $q = \frac{\ln(W)}{\ln(W)-1}$, and $q' = \frac{\ln(\tau)}{\ln(\tau)-1}$.
- Composite norm $\|(M, N, \beta, P)\| := \|M\|_{2,1} + \|N\|_{2,1} + \|\beta\|_1 + \sqrt{\sum_{j=1}^{\tau} \|P_j\|_F^2}$.
- Composite norm constraint $R_{\hat{\Theta}}$, and the corresponding set of pseudo-LDSs $\mathcal{K} = \{\hat{\Theta} : \|\hat{\Theta}\| \leq R_{\hat{\Theta}}\}$.

Crucially, $y(\hat{\Theta}; x_{1:t}, y_{t-1:t-d})$ is linear in each of $M, N, P, \beta$; consequently, the least-squares loss $\|y(\hat{\Theta}; x_{1:t}) - y\|^2$ is convex, and can be minimized in polynomial time. To this end, our online prediction algorithm is follow-the-regularized-leader (FTRL), which requires the solution of a convex program at each iteration. We choose this regularization to obtain the strongest theoretical guarantee, and provide a brief note in Section 5 on alternatives to address performance issues.

At a high level, our algorithm works by first approximating the response of an LDS by an autoregressive model of order $(\tau, \tau)$, then refining the approximation using wave-filters with a *phase* component. Specifically, the blocks of $M$ and $N$ corresponding to filter index $h$ and phase index $p$ specify the linear dependence of $y_t$ on a certain convolution of the input time series, whose kernel is the pointwise product of $\phi_h$ and a sinusoid with period $W/p$. The structural result which drives the theorem is that the dynamics of any true LDS are approximated by such a pseudo-LDS, with reasonably small parameters and coefficients.

Note that the autoregressive component in our definition of a pseudo-LDS is slightly more restricted than multivariate autoregressive models: the coefficients $\beta_j$ are *scalar*, rather than allowed to be

arbitrary matrices. These options are interchangeable for our purposes, without affecting the asymptotic regret; we choose to use scalar coefficients for a more streamlined analysis.

The online prediction algorithm is fully specified in Algorithm 1; the parameter choices that give the best asymptotic theoretical guarantees are specified in the appendix, while typical realistic settings are outlined in Section 5.

# 4   Analysis

There are three parts to the analysis, which we outline in the following subsections: proving the approximability of an LDS by a pseudo-LDS, bounding the regret incurred by the algorithm against the best pseudo-LDS, and finally analyzing the effect of noise $L$. The full proofs are in Appendices A, B, and C, respectively.

## 4.1   Approximation theorem for general LDSs

We develop a more general analogue of the structural result from [HSZ17], which holds for systems with asymmetric transition matrix $A$.

**Theorem 3** (Approximation theorem; informal). *Consider a noiseless LDS (given by (2) and (3) with $\eta_t, \xi_t = 0$) satisfying the assumptions in Section 2.1.*

*There is $k = O\left(\text{poly}\log\left(T, R_\Theta, R_\Psi, R_1, R_x, \frac{1}{\varepsilon}\right)\right)$, $W = O\left(\text{poly}(\tau, R_\Theta, R_\Psi, R_1, R_x, T)\right)$ and a pseudo-LDS $\hat{\Theta}$ of norm $O(\text{poly}(R_\Theta, R_\Psi, R_1, \tau, k))$ such that $\hat{\Theta}$ approximates $y_t$ to within $\varepsilon$ for $1 \leq t \leq T$:*

$$\left\| y(\hat{\Theta}; x_{1:t}, y_{t-1:t-\tau}) - y_t \right\| \leq \varepsilon. \tag{5}$$

For the formal statement (with precise bounds) and proof, see Appendix A.2. In this section we give some intuition for the conditions and an outline of the proof.

First, we explain the condition on the polynomial $p$. As we show in Appendix A.1 we can predict using a pure autoregressive model, without wavefilters, if we require $p$ to have all eigenvalues of $A$ as roots (i.e., it is divisible by the minimal polynomial of $A$). However, the coefficients of this polynomial could be very large. The size of these coefficients will appear in the bound for the main theorem, as using large coefficients in the predictor will make it sensitive to noise.

Requiring $p$ only to have the phases of eigenvalues of $A$ as roots can decrease the coefficients significantly. As an example, consider if $A$ has many $d/3$ distinct eigenvalues with phase 1, and similarly for $\omega$, and $\overline{\omega}$, and suppose their absolute values are close to 1. Then the minimal polynomial is approximately $(x-1)^{\frac{d}{3}}(x-\omega)^{\frac{d}{3}}(x-\overline{\omega})^{\frac{d}{3}}$ which can have coefficients as large as $\exp(\Omega(d))$. On the other hand, for the theorem we can take $p(x) = (x-1)(x-\omega)(x-\overline{\omega})$ which has degree 3 and coefficients bounded by a constant. Intuitively, the wavefilters help if there are few distinct phases, or they are well-separated (consider that if the phases were exactly the $d$th roots of unity, that $p$ can be taken to be $x^d - 1$). Note that when the roots are real, we can take $p = x - 1$ and the analysis reduces to that of [HSZ17].

We now sketch a proof of Theorem 3. Motivation is given by the Cayley-Hamilton Theorem, which says that if $p$ is the characteristic polynomial of $A$, then $p(A) = O$. This fact tells us that the $h_t = A^t h_0$ satisfies a linear recurrence of order $\tau = \deg p$: if $p(x) = x^\tau + \sum_{t=1}^\tau \beta_j x^{\tau-j}$, then $h_t + \sum_{t=1}^\tau \beta_j h_{t-j} = 0$.

If $p$ has only the phases as the roots, then $h_t + \sum_{t=1}^\tau \beta_j h_{t-j} \neq 0$ but can be written in terms of the wavefilters. Consider for simplicity the 1-dimensional (complex) LDS $y_t = \alpha y_{t-1} + x_t$, and let $\alpha = r\omega$ with $|\omega| = 1$. Suppose $p(x) = x^\tau + \sum_{t=1}^\tau \beta^j x^{\tau-j} = 0$ and $p(\omega) = 0$. In general the LDS is a "sum" of LDS's that are in this form. Summing the past $\tau$ terms with coefficients given by $\beta$,

$$
\begin{array}{llllll}
y_t = & x_t & +\alpha x_{t-1} & +\cdots & +\alpha^\tau x_{t-\tau} & +\cdots \\
+\beta_1(y_{t-1} = & & x_{t-1} & +\cdots & +\alpha^{\tau-1}x_{t-\tau} & +\cdots) \\
& \vdots & & \vdots & \vdots & \\
+\beta_\tau(y_{t-\tau} = & & & & x_{t-\tau} & +\cdots)
\end{array}
$$

The terms $x_t, \ldots, x_{t-\tau+1}$ can be taken care of by linear regression. Consider a term $x_j, j < t - \tau$ in this sum. The coefficient is $\alpha^{j-(t-\tau)}(\alpha^\tau + \beta_1 \alpha^{\tau-1} + \cdots + \beta_\tau)$. Because $p(\omega) = 0$, this can be written as

$$\alpha^{j-(t-\tau)}((\alpha^\tau - \omega^\tau) + \beta_1(\alpha^{\tau-1} - \omega^{\tau-1}) + \cdots). \tag{6}$$

Factoring out $1 - r$ from each of these terms show that $y_t + \beta_1 y_{t-1} + \cdots + \beta_\tau y_{t-\tau}$ can be expressed as a function of a convolution of the vector $((1-r)r^{t-1}\omega^{t-1})$ with $x_{1:T}$. The wavefilters were designed precisely to approximate the vector $\mu(r) = ((1-r)r^{t-1})_{1 \le t \le T}$ well, hence $y_t + \beta_1 y_{t-1} + \cdots + \beta_\tau y_{t-\tau}$ can be approximated using the wavefilters multiplied by phase and convolved with $x$. Note that the $1 - r$ is necessary in order to make the $L^2$ norm of $((1-r)r^{t-1})_{1 \le t \le T}$ bounded, and hence ensure the wavefilters have bounded coefficients.

## 4.2 Regret bound for pseudo-LDSs

As an intermediate step toward the main theorem, we show a regret bound on the total least-squares prediction error made by Algorithm 1, compared to the best *pseudo-LDS* in hindsight.

**Theorem 4** (FTRL regret bound; informal). *Let $\hat{y}_1^*, \ldots, \hat{y}_T^*$ denote the predictions made by the fixed pseudo-LDS minimizing the total squared-norm error. Then, there is a choice of parameters for which the decision set $\mathcal{K}$ contains all LDSs which obey the assumptions from Section 2.1, for which the predictions $\hat{y}_1, \ldots, \hat{y}_T$ made by Algorithm 1 satisfy*

$$\sum_{t=1}^T \|\hat{y}_t - y_t\|^2 - \sum_{t=1}^T \|\hat{y}_t^* - y_t\|^2 \le \tilde{O}\left(\text{poly}(R, d')\sqrt{T}\right).$$

*where $R_1, R_x, R_y, R_\Theta, R_\Psi \le R, m, n, d \le d'$.*

The regret bound follows by applying the standard regret bound of follow-the-regularized-leader (see, e.g. [Haz16]). However, special care must be taken to ensure that the gradient and diameter factors incur only a $\text{poly}\log(T)$ factor, noting that the discretization parameter $W$ (one of the dimensions of $M$ and $N$) must depend polynomially on $T/\varepsilon$ in order for the class of pseudo-LDSs to approximate true LDSs up to error $\varepsilon$. To this end, we use a modification of the strongly convex matrix regularizer found in [KSST12], resulting in a regret bound with logarithmic dependence on $W$.

Intuitively, this is possible due to the $d$-sparsity (and thus $\ell_1$ boundedness) of the phases of true LDSs, which transfers to an $\ell_1$ bound (in the phase dimension only) on pseudo-LDSs that compete with LDSs of the same size. This allows us to formulate a second convex relaxation, on top of that of wave-filtering, for simultaneous identification of eigenvalue phase and magnitude. For the complete theorem statement and proof, see Appendix B.

We note that the regret analysis can be used directly with the approximation result for autoregressive models (Theorem 1), without wave-filtering. This way, one can straightforwardly obtain a sublinear regret bound against autoregressive models with bounded coefficients. However, for the reasons discussed in Section 4.1, the wave-filtering technique affords us a much stronger end-to-end result.

## 4.3 Pseudo-LDSs compete with true LDSs

Theorem 3 shows that there exists a pseudo-LDS approximating the actual LDS to within $\varepsilon$ in the noiseless case. We next need to analyze the best approximation when there is noise. We show in Appendix C (Lemma 14) that if the noise is bounded $(\sum_{t=1}^T \|\eta_t\|_2^2 + \|\xi_t\|_2^2 \le L)$, we incur an additional term equal to the size of the perturbation $\sqrt{L}$ times a *competitive ratio* depending on the dynamical system, for a total of $R_\infty \tau^{\frac{3}{2}} R_\Theta R_\Psi \sqrt{L}$. We show this by showing that any noise has a bounded effect on the predictions of the pseudo-LDS.[2]

Letting $\hat{y}_t^*$ be the predictions of the best pseudo-LDS, we have

$$\sum_{t=1}^T \|\hat{y}_t - y_t\|_2^2 = \left(\sum_{t=1}^T \|\hat{y}_t - y_t\|_2^2 - \sum_{t=1}^T \|\hat{y}_t^* - \hat{y}_t\|_2^2\right) + \sum_{t=1}^T \|\hat{y}_t^* - \hat{y}_t\|_2^2. \tag{7}$$

The first term is the regret, bounded by Theorem 4 and the second term is bounded by the discussion above, giving the bound in the Theorem 1.

For the complete proof, see Appendix C.2.

## 5 Experiments

We exhibit two experiments on synthetic time series, which are generated by randomly-generated ill-conditioned LDSs. In both cases, $A \in \mathbb{R}^{10 \times 10}$ is a block-diagonal matrix, whose 2-by-2 blocks are rotation matrices $[\cos \theta \quad -\sin \theta; \sin \theta \quad \cos \theta]$ for phases $\theta$ drawn uniformly at random. This comprises a hard case for direct system identification: long-term time dependences between input and output, and the optimization landscape is non-convex, with many local minima. Here, $B \in \mathbb{R}^{10 \times 10}$ and $C \in \mathbb{R}^{2 \times 10}$ are random matrices of standard i.i.d. Gaussians. In the first experiment, the inputs $x_t$ are i.i.d. spherical Gaussians; in the second, the inputs are Gaussian block impulses.

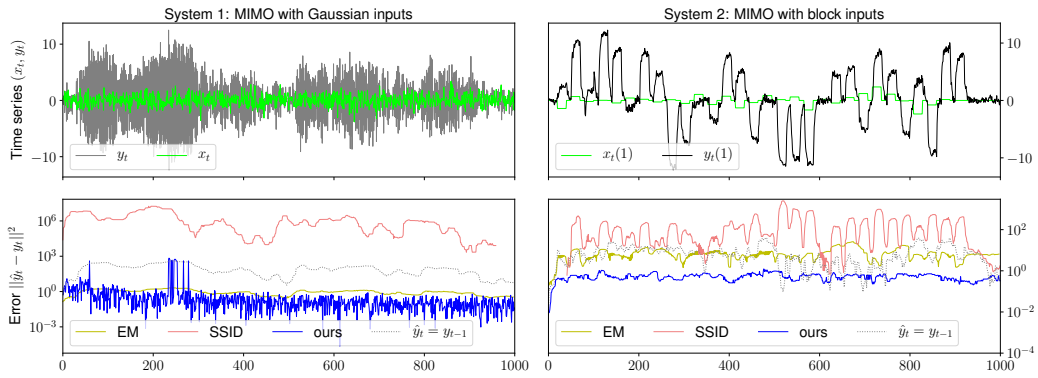

Figure 1: Performance of Algorithm 1 on synthetic 10-dimensional LDSs. For clarity, error plots are smoothed by a median filter. Blue = ours, yellow = EM, red = SSID, **black** = true responses, green = inputs, dotted lines = "guess the previous output" baseline. Horizontal axis is time. *Left:* Gaussian inputs; SSID fails to converge, while EM finds a local optimum. *Right:* Block impulse inputs; both baselines find local optima.

We make a few straightforward modifications to Algorithm 1, for practicality. First, we replace the scalar autoregressive parameters with matrices $\beta_j \in \mathbb{R}^{m \times m}$. Also, for performance reasons, we use ridge regularization instead of the prescribed pseudo-LDS regularizer with composite norm constraint. We choose an autoregressive parameter of $\tau = d = 10$ (in accordance with the theory), and $W = 100$.

As shown in Figure 1, our algorithm significantly outperforms the baseline methods of system identification followed by Kalman filtering. The EM and subspace identification (SSID; see [VODM12]) algorithms finds a local optimum; in the experiment with Gaussian inputs, the latter failed to converge (left).

We note that while the main online algorithm from [HSZ17], Algorithm 1 is significantly faster than baseline methods, ours is not. The reason is that we incur at least an extra factor of $W$ to compute and process the additional convolutions. To remove this phase discretization bottleneck, many heuristics are available for phase identification; see Chapter 6 of [Lju98].

## 6 Conclusion

We gave the first, to the best of our knowledge, polynomial-time algorithm for prediction in the general LDS setting without dependence on the spectral radius parameter of the underlying system. Our algorithm combines several techniques, namely the recently introduced wave-filtering method, as well as convex relaxation and linear filtering.

One important future direction is to improve the regret in the setting of (non-adversarial) Gaussian noise. In this setting, if the LDS is explicitly identified, the best predictor is the Kalman filter, which, when unrolled, depends on feedback for *all* previous time steps, and only incurs a cost $O(L)$ from noise in (4). It is of great theoretical and practical interest to compete directly with the Kalman filter without system identification.

## Footnotes

[1]Note that no bound on $\|y_t\|$ is required for the approximation theorem; $R_y$ only appears in the regret bound.

[2]In other words, the prediction error of the pseudo-LDS is stable to noise, and we bound its $H^\infty$ norm.

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
