[Supplementary Material]

# Supplementary Material for "Spectral Filtering for General Linear Dynamical Systems"

**Elad Hazan**
Princeton University & Google AI Princeton
ehazan@cs.princeton.edu

**Holden Lee**
Princeton University
holdenl@princeton.edu

**Karan Singh**
Princeton University & Google AI Princeton
karans@cs.princeton.edu

**Cyril Zhang**
Princeton University & Google AI Princeton
cyril.zhang@cs.princeton.edu

**Yi Zhang**
Princeton University & Google AI Princeton
y.zhang@cs.princeton.edu

## A   Proof of approximation theorem

Note that throughout the paper, we can assume without loss of generality that $D = 0$ (the zero matrix) for any LDS we wish to approximate. When this is not the case, we simply add an autoregressive component with delay zero $P_0$, which increases the norm of the pseudo-LDS by an additive $R_\Theta$. Also, we use $\iota$ to denote the imaginary unit (as opposed to index $i$).

### A.1   Warm-up: Simple autoregressive model

As a warm-up, we first establish rigorous regret bounds for an autoregressive model, depending on properties of the the minimal polynomial of the linear dynamical system. Next we will see how introducing wavefilters can improve these bounds.

**Theorem 1.** *Consider a noiseless LDS* $\Theta = (A, B, C, h_0 = 0)$, *where* $A = \Psi\Lambda\Psi^{-1}$ *is diagonalizable with spectral radius* $\leq 1$. *Let* $p(x)$ *be a monic polynomial of degree* $\tau$ *such that* $p(A) = 0$.[1] *Suppose* $\|p\|_1 \leq R_1$,[2] $\|B\|_2, \|C\|_2 \leq R_\Theta$, *and* $\|\Psi\|_F \|\Psi^{-1}\|_F \leq R_\Psi$.

*Suppose* $y_1, \ldots, y_t$ *is generated from the LDS with inputs* $x_t$. *Then there exist* $\beta \in \mathbb{R}^\tau$ *and* $P_0, \ldots, P_{\tau-1} \in \mathbb{R}^{m \times n}$ *satisfying*

$$\|\beta\|_1 \leq R_1 \tag{2}$$

$$\|P_j\|_F \leq R_1 R_\Theta^2 R_\Psi \tag{3}$$

*such that*

$$y_t = -\sum_{j=1}^{\tau} \beta_j y_{t-j} + \sum_{j=0}^{\tau-1} P_j x_{t-j}. \tag{4}$$

$$\|p\|_1 = \sum_{j=0}^{\tau} |\beta_j| \tag{1}$$

*Proof.* Unfolding the LDS,

$$y_t = \sum_{i=0}^{t-1} CA^i B x_{t-i} \tag{5}$$

$$y_{t-j} = \sum_{i=0}^{t-j-1} CA^i B x_{t-j-i} = \sum_{i=j}^{t-1} CA^{i-j} B x_{t-i}. \tag{6}$$

Let $p(x) = \sum_{j=0}^{\tau} \beta_j x^{\tau-j}$ (with $\beta_0 = 1$). Then

$$\sum_{j=0}^{\tau} \beta_j y_{t-j} \tag{7}$$

$$= \sum_{\substack{0 \le j \le \tau \\ j \le i \le t-1}} \beta_j CA^{i-j} B x_{t-i} \tag{8}$$

$$= \sum_{i=0}^{\tau-1} \sum_{j=0}^{i} \beta_j CA^{i-j} B x_{t-i} + C \underbrace{\sum_{i=\tau}^{t-1} \left( \sum_{j=0}^{\tau} \beta_j A^{i-j} \right) B x_{t-i}}_{=0} \tag{9}$$

using the fact that $\sum_{j=0}^{\tau} \beta_j A^j = A^{i-\tau} p(A) = 0$. Writing this in the autoregressive format,

$$y_t = -\sum_{j=1}^{\tau} \beta_j y_{t-j} + \sum_{i=0}^{\tau-1} \sum_{j=0}^{i} \beta_j CA^{i-j} B x_{t-i}. \tag{10}$$

We let $P_i = \sum_{j=0}^{i} \beta_j CA^{i-j} B$ for $0 \le i \le \tau - 1$ and check that this satisfies the conditions. Note $\|MN\|_F \le \min\{\|M\|_2 \|N\|_F, \|M\|_F \|N\|_2\}$. By the $L^1$-$L^\infty$ inequality,

$$\|P_i\|_F \tag{11}$$

$$= \left\| \sum_{j=0}^{i} \beta_j CA^{i-j} B \right\|_F \tag{12}$$

$$\le \left( \sum_{j=0}^{i} |\beta_j| \right) \max_{0 \le j \le i} \left\| CA^{i-j} B \right\|_F \tag{13}$$

$$\le R_1 \|C\|_2 \|\Psi\|_F \|\Lambda\|_2 \|\Psi^{-1}\|_F \|B\|_2 \tag{14}$$

$$\le R_1 R_\Theta^2 R_\Psi. \tag{15}$$

$\square$

## A.2 Autoregressive model with wavefilters: An approximate convex relaxation for asymmetric LDS

If we add wavefiltered inputs to the regression, the bounds depend not on the minimal polynomial $p$ of $A$, but rather on the minimal polynomial having all the $\omega_\ell$ as zeros, where $\omega_\ell$ are the phases of zeros of the characteristic polynomial of $A$. For example, if all the roots are real and distinct, then the polynomial is $x - 1$ rather than $\prod_{p(\alpha)=0}(x - \alpha)$. This can be an exponential improvement. For example, consider the case where all the $\alpha$ are close to 1. Then the minimal polynomial is close to $(x - 1)^d$, which has coefficients as large as $\exp(\Omega(d))$. Note the case of real roots reduces to [HSZ17].

First, we introduce some notation for convenience. Note that in Definition 2, $\sum_{u=1}^{T} \phi_h(u) x_{t-u}(i) \cos\left(\frac{2\pi up}{W}\right)$ is a certain convolution. For $a \in \mathbb{R}^T$ we define

$$a^{(\omega)} := (a_j \omega^j)_{1 \le j \le T} \tag{16}$$

$$a^{(\cos,\theta)} := (a_j \cos(j\theta))_{1 \le j \le T} \tag{17}$$

$$a^{(\sin,\theta)} := (a_j \sin(j\theta))_{1 \le j \le T}. \tag{18}$$

so that we can write Definition 2 as

$$y(M, N, \beta, P; x_{1:t}, y_{t-1:t-d}) \tag{19}$$

$$= \sum_{u=1}^{\tau} \beta_u y_{t-u} + \sum_{j=0}^{\tau-1} P_j x_{t-j} \tag{20}$$

$$+ \sum_{p=0}^{P-1} \sum_{h=1}^{k} \sum_{u=\tau}^{t} \left( M(p, h, :, :) \sigma_h^{\frac{1}{4}} (\phi_h^{(\cos, \frac{2\pi u p}{P})} * x)_{t-\tau} \right. \tag{21}$$

$$\left. + N(p, h, :, :) \sigma_h^{\frac{1}{4}} (\phi_h^{(\sin, \frac{2\pi u p}{P})} * x)_{t-\tau} \right) \tag{22}$$

We now give a more precise statement of Theorem 3 and prove it.

**Theorem 2.** *Consider a LDS $\Theta = (A, B, C, h_0 = 0)$, where $A$ is diagonalizable with spectral radius $\leq 1$. Let the eigenvalues of $A$ be $\alpha_\ell = \omega_\ell r_\ell$ for $1 \leq \ell \leq d$, where $|\omega_\ell| = 1$ and $r_\ell \in \mathbb{R}_{\geq 0}$. Let $S$ be the set of $\omega_\ell$. Let $p(x)$ be a monic polynomial of degree $\tau$ such that $p(\omega_\ell) = 0$ for each $\ell$. Suppose $\|p\|_1 \leq R_1$, $\|B\|_2, \|C\|_2 \leq R_\Theta$, and the diagonalization $A = \Psi \Lambda \Psi^{-1}$ satisfies $\|\Psi\|_F \|\Psi^{-1}\|_F \leq R_\Psi$.*

*Suppose $y_1, \ldots, y_T$ is generated from the LDS with inputs $x_1, \ldots, x_T$ such that $\|x_t\|_2 \leq R_x$ for all $1 \leq t \leq T$. Then there is $k = O\left(\ln T \ln\left(\frac{\tau R_\Theta R_\Psi R_1 R_x T}{\varepsilon}\right)\right)$, $W = O\left(\frac{\tau R_\Theta^2 R_\Psi R_1 R_x T^{\frac{3}{2}}}{\varepsilon}\right)$, $M, N \in \mathbb{R}^{W \times k \times n \times m}$, $\beta \in \mathbb{R}^\tau$ and $P_0, \ldots, P_{\tau-1} \in \mathbb{R}^{m \times n}$ satisfying*

$$\|\beta\|_1 \leq R_1 \tag{23}$$

$$\|P_j\|_F \leq R_1 R_\Theta^2 R_\Psi \tag{24}$$

$$\sqrt{\|M\|_{2,1}^2 + \|N\|_{2,1}^2} = O(R_\Theta^2 R_\Psi R_1 \tau k^{\frac{1}{2}}) \tag{25}$$

*such that the pseudo-LDS $(M, N, \beta, P)$ approximates $y_t$ within $\varepsilon$ for $1 \leq t \leq T$:*

$$\|y(M, N, \beta, P_{0:\tau-1}; x_{1:t}, y_{t-1:t-\tau}) - y_t\| \leq \varepsilon. \tag{26}$$

*Proof.* Our plan is as follows. First we show that choosing $\beta_j$ and $P_j$ as in Theorem 1,

$$\delta_t = y_t + \sum_{j=1}^{\tau} \beta_j y_{t-j} - \sum_{j=0}^{\tau-1} P_j x_{t-j} \tag{27}$$

can be approximated by

$$\delta_t^{(1)} := \sum_{\ell=1}^{d} \sum_{h=1}^{k} M_\ell'(h, :, :) \sigma_h^{\frac{1}{4}} (\phi_h^{(\omega_\ell)} * x)_{t-\tau} \tag{28}$$

for $M_\ell' \in \mathbb{C}^{k \times n \times m}$; this is obtained by a projection to the wavefilters. Next we approximate $\delta_t^{(1)}$ by discretization of the phase,

$$\delta_t^{(2)} := \sum_{\ell=1}^{d} \sum_{h=1}^{k} M_\ell'(h, :, :) \sigma_h^{\frac{1}{4}} (\phi_h^{(e^{2\pi \iota p_\ell / W})} * x)_{t-\tau} \tag{29}$$

for some integers $p_\ell \in [0, W-1]$. Finally, we show taking the real part gives something in the form

$$\delta_t^{(2)} = \sum_{\ell=1}^{d} \sum_{h=1}^{k} \left( M_\ell(h, :, :) \sigma_h^{\frac{1}{4}} (\phi_h^{(\cos, \frac{2\pi p_\ell}{W})} * x)_{t-\tau} \right. \tag{30}$$

$$\left. + N_\ell(h, :, :) \sigma_h^{\frac{1}{4}} (\phi_h^{(\sin, \frac{2\pi p_\ell}{W})} * x)_{t-\tau} \right) \tag{31}$$

This matches the form of a pseudo-LDS given in Definition 2 after collecting terms,

$$M(p,:,:,:) = \sum_{\ell:p_\ell=p} M_\ell \qquad\qquad N(p,:,:,:) = \sum_{\ell:p_\ell=p} N_\ell. \tag{32}$$

Now we carry out the plan. Again we have (9) except that this time the second term is not 0. Let $A = \Psi\Lambda\Psi^{-1}$, $v_\ell$ be the columns of $\Psi$ and $w_\ell^*$ be the rows of $\Psi^{-1}$. By assumption on $p$, $\sum_{j=0}^{\tau} \beta_j \omega_\ell^{\tau-j} = 0$ for each $\ell$, so the second term of (9) equals

$$\delta_t \tag{33}$$

$$= C \sum_{i=\tau}^{t-1} \left( \sum_{\ell=1}^{d} \sum_{j=0}^{\tau} \beta_j \alpha_\ell^{i-\tau} \alpha_\ell^{\tau-j} v_\ell w_\ell^* \right) B x_{t-i} \tag{34}$$

$$= C \sum_{i=\tau}^{t-1} \left( \sum_{\ell=1}^{d} \alpha_\ell^{i-\tau} \sum_{j=0}^{\tau} \beta_j (\alpha_\ell^{\tau-j} - \omega_\ell^{\tau-j}) v_\ell w_\ell^* \right) B x_{t-i} \tag{35}$$

$$= \sum_{\ell=1}^{d} C v_\ell w_\ell^* B \sum_{i=\tau}^{t-1} r_\ell^{i-\tau} \omega_\ell^{i-\tau} \sum_{j=0}^{\tau} \beta_j \omega_\ell^{\tau-j} (r_\ell^{\tau-j} - 1) x_{t-i} \tag{36}$$

$$= \sum_{\ell=1}^{d} C v_\ell w_\ell^* B \sum_{j=0}^{\tau} \sum_{i=\tau}^{t-1} \beta_j \left( r_\ell^{\tau-j} - 1 \right) r_\ell^{i-\tau} \omega_\ell^{i-j} x_{t-i} \tag{37}$$

$$= \sum_{\ell=1}^{d} C v_\ell w_\ell^* B \sum_{j=0}^{\tau} \beta_j \omega^{\tau-j} \frac{r_\ell^{\tau-j} - 1}{1 - r_\ell} \left( \mu(r_\ell)^{(\omega_\ell)} * x \right)_{t-\tau}. \tag{38}$$

Thus this equals

$$\delta_t \tag{39}$$

$$= \underbrace{\sum_{\ell=1}^{d} C v_\ell w_\ell^* B \sum_{j=0}^{\tau} \beta_j \omega_\ell^{\tau-j} \frac{r_\ell^{\tau-j} - 1}{1 - r_\ell} \left( \widetilde{\mu}(r_\ell)^{(\omega_\ell)} * x \right)_{t-\tau}}_{\delta_t^{(1)}} \tag{40}$$

$$+ \sum_{\ell=1}^{d} C v_\ell w_\ell^* B \sum_{j=0}^{\tau} \beta_j \omega_\ell^{\tau-j} \frac{r_\ell^{\tau-j} - 1}{1 - r_\ell} \tag{41}$$

$$\left( (\mu(r_\ell) - \widetilde{\mu}(r_\ell))^{(\omega_\ell)} * x \right)_{t-\tau} \tag{42}$$

where $\widetilde{\mu}_k(r)$ is the projection of $\mu_k(r)$ to the subspace spanned by the rows of $\phi_1, \ldots, \phi_k$. Let $\Phi$ be the matrix with rows $\phi_j$, let $D = \mathrm{diag}((\sigma_h^{\frac{1}{4}})_{1\le h\le k})$ and let $m_{\ell,h} \in \mathbb{R}^k$ be such that $\widetilde{\mu}(r_\ell) = \Phi D m_\ell = \sum_{h=1}^{k} m_{\ell,h} \sigma_h^{\frac{1}{4}} \phi_h$, i.e., $m_\ell = \sigma_h^{-\frac{1}{4}} \langle \phi_h, \mu(r_\ell) \rangle$. The purpose of $D$ is to scale down $\phi_h * x$ so that gradients will be small in the optimization analysis.

We show that the term (40) equals $\delta_t^{(1)}$ with some choice of $M_\ell'$.

$$(40) \tag{43}$$

$$= \sum_{\ell=1}^{d} C v_\ell w_\ell^* B \sum_{j=0}^{\tau-1} \beta_j \omega_\ell^{\tau-j} \frac{r_\ell^{\tau-j} - 1}{1 - r_\ell} \tag{44}$$

$$\sum_{h=1}^{k} m_{\ell,h} \sigma_h^{\frac{1}{4}} (\phi_h^{(\omega_\ell)} * x)_{t-\tau} \tag{45}$$

$$= \sum_{\ell=1}^{d} \sum_{h=1}^{k} M_\ell'(h,:,:) \sigma_h^{\frac{1}{4}} (\phi_h^{(\omega_\ell)} * x)_{t-\tau} \tag{46}$$

$$=\delta_t^{(1)} \tag{47}$$

where $M_\ell'(h, r, i) := (L_\ell)_{ri} m_{\ell,h}$ and

$$L_\ell = C v_\ell w_\ell^* B \sum_{j=0}^{\tau-1} \beta_j \omega_\ell^{\tau-j} \left( \frac{r_\ell^{\tau-j} - 1}{1 - r_\ell} \right) \tag{48}$$

We calculate the error. Here $\|x\|_\infty$ denotes $\max_i \|x_i\|_2$.

$$\sum_{\ell=1}^d \|L_\ell\|_F \tag{49}$$

$$= \sum_{\ell=1}^d \left\| C v_\ell w_\ell^* B \sum_{j=0}^{\tau-1} \beta_j \omega_\ell^{\tau-j} \frac{r_\ell^{\tau-j} - 1}{1 - r_\ell} \right\|_F \tag{50}$$

$$\leq \sum_{\ell=1}^d \|C\|_F \|B\|_F \|v_\ell\|_2 \|w_\ell\|_2 \|\beta\|_1 \tau \tag{51}$$

$$\leq \|C\|_F \|B\|_F \sqrt{\sum_{\ell=1}^d \|v_\ell\|_2^2 \sum_{\ell=1}^d \|w_\ell\|_2^2} \|\beta\|_1 \tau \tag{52}$$

$$\leq R_\Theta^2 R_\Psi R_1 \tau \tag{53}$$

$$\left\| \delta_t - \delta_t^{(1)} \right\|_2 \tag{54}$$

$$\leq \sum_{\ell=1}^d \|L_\ell\|_2 \max_\ell ((\mu(r_\ell) - \widetilde{\mu}(r_\ell))^{(\omega_\ell)} * x)_{t-\tau} \tag{55}$$

$$\leq R_\Theta^2 R_\Psi R_1 \tau \max_\ell \|\mu(r_\ell) - \widetilde{\mu}(r_\ell)\|_1 \|x\|_\infty \tag{56}$$

$$\leq R_\Theta^2 R_\Psi R_1 \tau \sqrt{T} \max_\ell \|\mu(r_\ell) - \widetilde{\mu}(r_\ell)\|_2 R_x \tag{57}$$

$$\leq R_\Theta^2 R_\Psi R_1 \tau \sqrt{T} c_1^{-k/\ln T} (\ln T)^{\frac{1}{4}}. \tag{58}$$

for some constant $c_1 < 1$, where the bound on $\|\mu(r_\ell) - \widetilde{\mu}(r_\ell)\|$ comes from Lemma C.2 in [HSZ17]. Choose $k = C \ln T \ln \left( \frac{\tau R_\Theta R_\Psi R_1 R_x T}{\varepsilon} \right)$ for large enough $C$ makes this $\leq \frac{\varepsilon}{2}$.

Now we analyze the effect of discretization. Write $\omega_\ell = e^{\theta_\ell \iota}$ where $\theta_\ell \in [0, 2\pi)$, and let $p_\ell$ be such that $\left| \theta_\ell - \frac{2\pi p_\ell}{W} \right| \leq \frac{\pi}{W}$ where the angles are compared modulo $2\pi$. Let $\omega_\ell' = e^{\frac{2\pi p_\ell \iota}{W}}$. We approximate $\delta_t^{(1)}$ with

$$\delta_t^{(2)} = \sum_{\ell=1}^d M_\ell'(h, :, :) \sigma_h^{\frac{1}{4}} (\phi_h^{(\omega_\ell')} * x)_{t-\tau} \tag{59}$$

Note that $|\omega_\ell^j - \omega_\ell'^j| \leq j \left| \theta_\ell - \frac{2\pi p_\ell}{W} \right| \leq \frac{\pi j}{W}$. We calculate the error. First note

$$\sum_{\ell=1}^d \sum_{h=1}^k \|M_\ell'(h, :, :)\|_F \sigma_h^{\frac{1}{4}} \leq \sum_{\ell=1}^d \|L_\ell\|_2 \max_{\ell,h} |\langle \phi_h, \mu(r_\ell) \rangle| \tag{60}$$

$$\leq R_\Theta^2 R_\Psi R_1 \tau \tag{61}$$

by (53) and $\|\mu(r_\ell)\|_2 \leq 1$. Then

$$\left\| \delta_t^{(1)} - \delta_t^{(2)} \right\|_2 \tag{62}$$

$$\leq \sum_{h=1}^k \left\| \sum_{\ell=1}^d M_\ell'(h, i, :)((\phi_h^{(\omega_\ell)} - \phi_h^{(\omega_\ell')}) * x)_{t-\tau}(i) \right\|_2 \tag{63}$$

$$\leq \left( \sum_{\ell=1}^{d} \sum_{h=1}^{k} \| M'_\ell(h,:,:) \|_F \, \sigma_h^{\frac{1}{4}} \right) \max_{h,\ell} \left\| \phi_h^{(\omega_\ell)} - \phi_h^{(\omega'_\ell)} \right\|_1 R_x \tag{64}$$

$$\leq R_\Theta^2 R_\Psi R_1 \tau \frac{\pi T}{W} \max_h \| \phi_h \|_1 R_x \tag{65}$$

$$\leq R_\Theta^2 R_\Psi R_1 \tau \frac{\pi T}{W} \sqrt{T} R_x \tag{66}$$

$$= \frac{R_\Theta^2 R_\Psi R_1 \tau \pi T^{\frac{3}{2}} R_x}{W} \leq \frac{\varepsilon}{2} \tag{67}$$

for $W = \frac{2\pi\tau R_\Theta^2 R_\Psi R_1 R_x T^{\frac{3}{2}}}{\varepsilon}$. This means $\left\| \delta_t - \delta_t^{(2)} \right\| \leq \varepsilon$.

Finally, we take the real part of $\delta_t^{(2)}$ (which doesn't increase the error, because $\delta_t$ is real) to get

$$\sum_{\ell=1}^{d} \sum_{i=1}^{n} \sum_{h=1}^{k} \sum_{u=1}^{T} \sigma_h^{\frac{1}{4}} \phi_h(u) x_{t-\tau-u}(i) \Re[M'_\ell(h,:,i) {\omega'_\ell}^u] \tag{68}$$

$$= \sum_{\ell=1}^{d} \sum_{i=1}^{n} \sum_{h=1}^{k} \sum_{u=1}^{T} \sigma_h^{\frac{1}{4}} \phi_h(u) x_{t-\tau-u}(i) \bigg( \Re(M'_\ell(h,:,i)) \tag{69}$$

$$\cos\left( \frac{2\pi p_\ell u}{W} \right) - \Im(M'_\ell(h,:,i)) \sin\left( \frac{2\pi p_\ell u}{W} \right) \bigg) \tag{70}$$

which is in the form (31) with $M_\ell = \Re(M'_\ell)$ and $N_\ell = -\Im(M'_\ell)$.

The bound on $\| M \|_{2,1}$ and $\| N \|_{2,1}$ is

$$\sqrt{ \| M \|_{2,1}^2 + \| N \|_{2,1}^2 } \tag{71}$$

$$\leq \sum_{\ell=1}^{d} \sqrt{ \| M_\ell \|_F^2 + \| N_\ell \|_F^2 } = \sum_{\ell=1}^{d} \| M'_\ell \|_F \tag{72}$$

$$= \sum_{\ell=1}^{d} \sqrt{ \sum_{h=1}^{k} \| M'_\ell(h,:,:) \|_F^2 } \tag{73}$$

$$= \sum_{\ell=1}^{d} \| L_\ell \|_F \| m_\ell \|_2 \tag{74}$$

$$= O(R_\Theta^2 R_\Psi^2 R_1 \tau k^{\frac{1}{2}}) \tag{75}$$

because

$$\| m_\ell \|_2 = \left( \sum_{h=1}^{k} \sigma_h^{-\frac{1}{2}} \langle \phi_h, \mu(r_\ell) \rangle^2 \right)^{\frac{1}{2}} \tag{76}$$

$$\leq \left( \sum_{h=1}^{k} O(1) \right)^{\frac{1}{2}} = O(k^{\frac{1}{2}}) \tag{77}$$

by Lemma E.4 in [HSZ17]. $\qquad\square$

## B Proof of the regret bound

In this section, we prove the following theorem:

**Theorem 3.** *Let $y_1^*, \ldots, y_T^*$ denote the predictions made by the fixed pseudo-LDS which has the smallest total squared-norm error in hindsight. Then, there is a choice of parameters for which the*

*decision set $\mathcal{K}$ contains all LDSs which obey the assumptions from Section 2.1, and Algorithm 1 makes predictions $\hat{y}_t$ such that*

$$\sum_{t=1}^{T} \|\hat{y}_t - y_t\|^2 - \sum_{t=1}^{T} \|\hat{y}_t^* - y_t\|^2 \ \leq \ \tilde{O}\left(Rd^{5/2}n \log^7 T \sqrt{T}\right),$$

*where the $\tilde{O}(\cdot)$ only suppresses factors polylogarithmic in $n, m, d, R_\Theta, R_x, R_y$, and $R_1^3 R_x^2 R_\Theta^4 R_\Psi^2 R_y^2 \ \leq \ R$.*

## B.1 Online learning with composite strongly convex regularizers

Algorithm 1 runs the follow-the-regularized-leader (FTRL) algorithm with the regularization

$$R(M, N, \beta, P) := \|M\|_{2,q}^2 + \|N\|_{2,q}^2 + \|\beta\|_{q'}^2 + \sum_{j=1}^{\tau} \|P_j\|_F^2,$$

where $q = \frac{\ln(W)}{\ln(W)-1}$ and $q' = \frac{\ln(\tau)}{\ln(\tau)-1}$. To achieve the desired regret bound, we need to show that this regularizer is strongly convex with respect to the composite norm considered in the algorithm. We will work with the following definition of strong convexity with respect to a norm:

**Definition 4** (Strong convexity w.r.t. a norm). A differentiable convex function $f : \mathcal{K} \to \mathbb{R}$ is $\alpha$-strongly convex with respect to the norm $\| \cdot \|$ if, for all $x, x + h \in \mathcal{K}$, it holds that

$$f(x + h) \ \geq \ f(x) + \langle \nabla f(x), h \rangle + \frac{\alpha}{2} \|h\|^2.$$

We first verify the following claim:

**Lemma 5.** *Suppose the convex functions $R_1, \ldots, R_n$, defined on domains $\mathcal{X}_1, \ldots, \mathcal{X}_n$, are $\alpha$-strongly convex with respect to the norms $\| \cdot \|_1, \ldots, \| \cdot \|_n$, respectively. Then, the function $R(x_1, \ldots, x_n) = \sum_{i=1}^{n} R_i(x_i)$, defined on the Cartesian product of domains, is $(\alpha/n)$-strongly convex w.r.t. the norm $\|(x_1, \ldots, x_n)\| = \sum_{i=1}^{n} \|x_i\|_i$.*

*Proof.* Summing the definitions of strong convexity, we get

$$R(x_1 + h_1, \ldots, x_n + h_n)$$

$$\geq \sum_{i=1}^{n} \langle \nabla R(x_i), h_i \rangle + \frac{\alpha}{2} \sum_{i=1}^{n} \|h_i\|_i^2$$

$$= \langle \nabla R(x_1, \ldots, x_n), \text{vec}(h_{1:n}) \rangle + \frac{\alpha}{2} \sum_{i=1}^{n} \|h_i\|_i^2$$

$$\geq \langle \nabla R(x_1, \ldots, x_n), \text{vec}(h_{1:n}) \rangle + \frac{\alpha}{2n} \left(\sum_{i=1}^{n} \|h_i\|_i\right)^2,$$

where the last inequality uses the AM-QM inequality. $\square$

Indeed, each term in $R(\hat{\Theta})$ is strongly convex in the respective term in the composite norm $\|\hat{\Theta}\|$. For $\|M\|_{2,q}^2$ (and, identically, $\|N\|_{2,q}^2$), we use Corollary 13 from [KSST12]:

**Lemma 6.** *The function $M \mapsto \|M\|_{2,q}^2$ is $\frac{1}{3\ln(W)}$-strongly convex w.r.t. the norm $\| \cdot \|_{2,1}$.*

This is a more general case of the fact that $\beta \mapsto \|\beta\|_{q'}^2$ is $\frac{1}{3\ln(\tau)}$-strongly convex w.r.t. the norm $\|\beta\|_1$. Finally, $\sum_{j=1}^{\tau} \|P_j\|_F^2$ is the squared Euclidean norm of $\text{vec}(P_{1:\tau})$, which is clearly 1-strongly convex w.r.t. the same Euclidean norm. Thus, applying Lemma 5, we have:

**Corollary 7.** *$R(\hat{\Theta})$ is $\alpha$-strongly convex in $\|\hat{\Theta}\|$, where $\alpha = \frac{1}{12\ln(\max(\tau,P))}$.*

Finally, we note an elementary upper bound for the dual of the norm $\|\hat{\Theta}\|$ in terms of the duals of the summands, which follows from the definition of dual norm:

**Lemma 8.** *For any norm $\| \cdot \|$, let $\| \cdot \|^*$ denote its dual norm $\|v\|^* = \sup_{\|w\| \leq 1}\langle v, w\rangle$. Then, if $\|(x_1, \ldots, x_n)\| = \sum_{i=1}^n \|x_i\|_i$, we have*

$$\|(x_1, \ldots, x_n)\|^* \leq \sum_{i=1}^n \|x_i\|_i^*.$$

**Corollary 9.** *In particular, for the norm $\|\hat{\Theta}\|$ we have defined on pseudo-LDSs, we have:*

$$\|(M, N, \beta, P)\|^* \tag{78}$$

$$\leq \|M\|_{2,\infty} + \|N\|_{2,\infty} + \|\beta\|_\infty + \sqrt{\sum_{j=1}^\tau \|P_j\|_F}. \tag{79}$$

## B.2 Regret of the FTRL algorithm

We state the standard regret bound of FTRL with a regularizer $R$ which is strongly convex with respect to an arbitrary norm $\| \cdot \|$. For a reference and proof, see Theorem 2.15 in [SS$^+$12].

**Lemma 10.** *In the standard online convex optimization setting with decision set $\mathcal{K}$, with convex loss functions $f_1, \ldots, f_T$, let $x_{1:t}$ denote the decisions made by the FTRL algorithm, which plays an arbitrary $x_1$, then $x_{t+1} := \arg\min_x \sum_{u=1}^t f_t(x) + \frac{R(x)}{\eta}$. Then, if $R(x)$ is $\alpha$-strongly convex w.r.t. the norm $\| \cdot \|$, we have the regret bound*

$$\text{Regret} := \sum_{t=1}^T f_t(x_t) - \min_x \sum_{t=1}^T f_t(x) \leq \frac{2R_{\max}}{\eta} + \frac{\eta T}{\alpha}G_{\max}^2,$$

*where $R_{\max} = \sup_{x \in \mathcal{K}} R(x)$ and $G_{\max} = \sup_{x \in \mathcal{K}} \|\nabla f(x)\|^*$.*

To optimize the bound, choose $\eta = \frac{\sqrt{2R_{\max}T/\alpha}}{G_{\max}}$, for a regret bound of

$$\text{Regret} \leq O\left(G_{\max}\sqrt{R_{\max}T/\alpha}\right).$$

With the facts established in Section B.1, this gives us the following regret bound, which gives an additive guarantee versus the best pseudo-LDS in hindsight:

**Corollary 11.** *For the sequence of squared-loss functions on the predictions $f_1, \ldots, f_T : \mathcal{K} \to \mathbb{R}$, Algorithm 1 produces a sequence of pseudo-LDSs $\hat{\Theta}_1, \ldots, \hat{\Theta}_T$ such that*

$$\sum_{t=1}^T f_t(\hat{\Theta}_t) - \min_{\hat{\Theta} \in \mathcal{K}} \sum_{t=1}^T f_t(\hat{\Theta}) \leq O\left(GR_{\hat{\Theta}}\sqrt{T\log\max(P, \tau)}\right),$$

*where $G$ is an upper bound on the quantity*

$$\|\nabla_M f_t(\hat{\Theta})\|_{2,\infty} + \|\nabla_N f_t(\hat{\Theta})\|_{2,\infty} \tag{80}$$

$$+ \|\nabla_\beta f_t(\hat{\Theta})\|_\infty + \sum_{j=1}^\tau \|\nabla_P f_t(\hat{\Theta})\|_F. \tag{81}$$

*(Here, $\nabla_M$ denotes the 4-tensor of partial derivatives with respect to the entries of $M$, and so on.)*

It suffices to establish an upper bound $R_{\hat{\Theta}}$ on the norm of a pseudo-LDS required to approximate a true LDS, as well as the gradient of the loss function in each of these dual norms. We can obtain the appropriate diameter constraint from Theorem 3:

$$R_{\hat{\Theta}} = \Theta\left(R_\Theta^2 R_\Psi R_1 \tau \sqrt{k}\right).$$

We bound $G$ in the following section.

## B.3 Bounding the gradient

In this section, we compute each gradient, and bound its appropriate norm.

**Lemma 12.** *Let $G$ be the bound in the statement in 11. Then,*

$$G \leq O\left(R_1^2 R_x^2 R_\Theta^2 R_\Psi R_y^2 \tau^{3/2} n k^{3/2} \log^2 T\right).$$

*Proof.* First, we use the result from Lemma E.5 from [HSZ17], which states that the $\ell_1$ norm of $\phi_h$ is bounded by $O(\log T/\sigma_h^{1/4})$, to bound the size of the convolutions taken by the algorithm:

**Lemma 13.** *For any $h \in [1,k]$ and $p \in [0, W-1]$, we have*

$$\|\sum_{u=1}^{T} \phi_h(u)\sigma_h^{1/4}\cos(2\pi up/W)x_{t-u}\|_2 \leq O\left(R_x\sqrt{n}\log T\right).$$

*The same holds upon replacing $\cos(\cdot)$ with $\sin(\cdot)$.*

*Proof.* We have that for each coordinate $i \in [n]$,

$$|\sum_{u=1}^{T} \phi_h(u)\sigma_h^{1/4}\cos(2\pi up/W)x_{t-u}(i)| \tag{82}$$

$$\leq \sum_{u=1}^{T} \phi_h(u)\sigma_h^{1/4}R_x \leq \|\phi_h\|_1 \sigma_h^{1/4}R_x \tag{83}$$

$$\leq O\left(R_x \log T\right). \tag{84}$$

$\square$

It will be useful to record an upper bound for the norm of the prediction residual $y(\hat{\Theta}) - y_t$, which appears in the gradient of the least-squares loss. By assumption, we have $\|y_t\|_2 \leq R_y$. By the constraint on $\|\Theta\|$ from the algorithm, and noting that $y(\hat{\Theta})$ a sum of matrix products of $M(p, h, :, :)$ and convolutions of the form of the LHS in Lemma 13, we can obtain a bound on $y(\hat{\Theta})$ as well:

$$\|y(\hat{\Theta}) - y_t\|_2 \leq O\left(R_y\|y(\hat{\Theta})\|_2\right) \tag{85}$$

$$\leq O\left(R_y R_{\hat{\Theta}} \cdot (\sqrt{nk}\log T R_x + R_x + R_y)\right) \tag{86}$$

$$\leq O\left(R_1^2 R_x R_\Theta^2 R_\Psi R_y^2 \tau \sqrt{nk} \log T\right). \tag{87}$$

Call this upper bound $U$ for short. First, we compute the gradients with respect to the 4-tensors $M$ and $N$. Fixing one phase $p$ and filter index $k$, we have:

$$\nabla_M f_t(\hat{\Theta})(p, h, :, :) \tag{88}$$

$$=2\left(y(\hat{\Theta}) - y_t\right)\left(\sum_{u=1}^{T} \phi_h(u)\cos(2\pi up/W)x_{t-u}\right)^\top, \tag{89}$$

so that

$$\|\nabla_M f_t(\hat{\Theta})(p, h, :, :)\|_F^2 \tag{90}$$

$$\leq 4\|y(\hat{\Theta}) - y_t\|^2 \|\sum_{u=1}^{T} \phi_h(u)\cos(2\pi up/W)x_{t-u}\|^2 \tag{91}$$

$$\leq U \cdot O\left(R_x\sqrt{n}\log T\right) \tag{92}$$

Thus, we have

$$\|\nabla_M f_t(\hat{\Theta})\|_{2,\infty} = \max_p \sqrt{\sum_{h=1}^{k} \|\nabla_M f_t(\hat{\Theta})(p, h, :, :)\|_F^2} \tag{93}$$

$$\leq U \cdot O\left(R_x\sqrt{nk}\log T\right) \tag{94}$$

The same bound holds for $\|\nabla_N f_t(\hat{\Theta})\|_{2,\infty}$.

For the $\beta$ part of the gradient, we have

$$\nabla_\beta f_t(\hat{\Theta})(j) = 2\left(y(\hat{\Theta}) - y_t\right)^\top y_{t-j}, \tag{95}$$

so that we have an entrywise bound of

$$\|\nabla_\beta f_t(\hat{\Theta})\|_\infty \leq O\left(R_y \cdot U\right). \tag{96}$$

Finally, for the $P_j$ part, we have

$$\nabla_{P_j} f_t(\hat{\Theta}) = 2\left(y(\hat{\Theta}) - y_t\right) x_{t-j}^\top, \tag{97}$$

so that

$$\sqrt{\sum_{j=1}^\tau \|\nabla_{P_j} f_t(\hat{\Theta})\|_F^2} \leq O\left(\sqrt{\tau} R_x \cdot U\right). \tag{98}$$

The claimed bound on $G$ follows by adding these bounds. □

The final regret bound follows by combining Lemma 12 and Corollary 11, with the choices $k = \Theta\left(\log T \log\left(\frac{\tau R_\Theta R_\Psi R_1 R_x T}{\varepsilon}\right)\right)$, $\tau = \Theta(d)$, $W = \Theta\left(\tau R_\Theta^2 R_\Psi R_1 R_x T^3\right)$.

## C Proof of main theorem: Competitive ratio bounds

### C.1 Perturbation analysis

To prove the main theorem, we first need to analyze the approximation when there is noise. Compared to the noiseless case, as analyzed in Theorem 2, we incur an additional term equal to the size of the perturbation times a *competitive ratio* depending on the dynamical system.

**Lemma 14.** *Consider an LDS $\Theta = (A, B, C, h_0 = 0)$ that satisfies the conditions of Theorem 3. Consider the LDS under adversarial noise* (2)–(3). *Let $y_t(x_{1:T}, \eta_{1:T}, \xi_{1:T})$ be the output at time $t$ given inputs $x_{1:T}$ and noise $\eta_{1:T}, \xi_{1:T}$. Let $\hat{y}_t(x_{1:T}, \eta_{1:T}, \xi_{1:T})$ be the prediction made by the pseudo-LDS at time $t$. Suppose that $(M, N, -\beta, P)$ is a pseudo-LDS that predicts well when there is no noise:*

$$\|\hat{y}_{1:T}(x_{1:T}, 0, 0) - y_{1:T}(x_{1:T}, 0, 0)\|_2 \tag{99}$$

$$= \left(\sum_{t=1}^T \|\hat{y}_t(x_{1:T}, 0, 0) - y_t^*(x_{1:T}, 0, 0)\|_2^2\right)^{\frac{1}{2}} \leq \varepsilon. \tag{100}$$

*For bounded adversarial noise $\sum_{t=1}^T \|\eta_t\|^2 + \|\xi_t\|^2 \leq L$,*

$$\|\hat{y}_{1:T}(x_{1:T}, \eta_{1:T}, \xi_{1:T}) - y_{1:T}(x_{1:T}, \eta_{1:T}, \xi_{1:T})\|_2 \tag{101}$$

$$\leq \varepsilon + O(\|\beta\|_\infty \tau^{\frac{3}{2}} R_\Theta R_\Psi \sqrt{L}). \tag{102}$$

*Note that the initial hidden state can be dealt with by considering it as noise in the first step $\eta_1$.*

*Proof.* Note that $\hat{y}_t$ is a linear function of $x_{1:T}, \eta_{1:T}, \xi_{1:T}$.

$$\hat{y}_t(x_{1:T}, \eta_{1:T}, \xi_{1:T}) - y_t \tag{103}$$

$$= [\hat{y}_t(0, 0, \xi_{1:T}) - y_t(0, 0, \xi_{1:T})] \tag{104}$$

$$+ [\hat{y}_t(0, \eta_{1:T}, 0) - y_t(0, \eta_{1:T}, 0)] \tag{105}$$

$$+ [\hat{y}_t(x_{1:T}, 0, 0) - y_t(x_{1:T}, 0, 0)]. \tag{106}$$

This says that the residual is the sum of the residuals incurred by each $\xi_t$ and $\eta_t$ individually, plus the residual for the non-noisy LDS, $\hat{y}_t(x_{1:T}, 0, 0) - y_t(x_{1:T}, 0, 0)$.

We first analyze the effect of a single perturbation to the observation $\xi_t$. Suppose $\eta_t = 0$ for all $t$ and $\xi_t = 0$ except for $t = u$ (so that $y_t = 0$ for all $t$ except $t = u$, where $y_u = \xi_u$). The predictions $\hat{y}_t$ are zero when $t \notin [u, u+\tau]$, because then the prediction does not depend on $y_u$. For $u \leq t \leq u+\tau$,

$$\|\hat{y}_t(0, 0, \xi_{1:T}) - y_t(0, 0, \xi_{1:T})\|_2 \tag{107}$$

$$= \left\| y_t + \sum_{j=1}^{\tau} \beta_j y_{t-j} \right\|_2 \tag{108}$$

$$\leq |\beta_{t-u}| \, \|\xi_u\|_2 \tag{109}$$

$$\|\hat{y}_{1:T}(0, 0, \xi_{1:T}) - y_{1:T}(0, 0, \xi_{1:T})\|_2 \tag{110}$$

$$\leq \left( \sum_{t=u}^{u+\tau} |\beta_{t-u}|^2 \right)^{\frac{1}{2}} \|\xi_u\|_2 \tag{111}$$

$$= \|\beta\|_2 \, \|\xi_u\|_2 \, . \tag{112}$$

Now we analyze the effect of a single perturbation to the hidden state $\eta_t$. Suppose $\xi_t = 0$ for all $t$ and $\eta_t = 0$ except for $t = u$ (so that $h_t = 0$ for all $t < u$, $h_u = \eta_u$, and the system thereafter evolves according to the LDS). For simplicity, we may as well consider the case where $u = 1$, i.e., the perturbation is to the initial hidden state. When $t \leq \tau$, the error is bounded by

$$\|\hat{y}_t(0, \eta_{1:T}, 0) - y_t(0, \eta_{1:T}, 0)\|_2 \tag{113}$$

$$= \left\| y_t + \sum_{j=1}^{t-1} \beta_j y_{t-j} \right\|_2 = \left\| C \sum_{j=0}^{t-1} \beta_j A^{t-j-1} \xi_1 \right\| \tag{114}$$

$$\leq \|C\|_2 \, \|\beta\|_1 \, \|\Psi\|_F \, \|\Lambda\|_2 \, \|\Psi^{-1}\|_F \, \|\eta_1\|_2 \tag{115}$$

$$\leq R_\Theta R_\Psi \, \|\beta\|_1 \, \|\eta_1\|_2 \tag{116}$$

$$\|\hat{y}_{1:\tau}(0, \eta_{1:T}, 0) - \hat{y}_{1:\tau}(0, \eta_{1:T}, 0)\|_2 \tag{117}$$

$$\leq \tau^{\frac{1}{2}} R_\Theta R_\Psi \, \|\beta\|_1 \, \|\eta_1\|_2 \, . \tag{118}$$

When $t > \tau$, the error is (let $\beta_0 = 1$)

$$y_t(0, \eta_{1:T}, 0) - \hat{y}_t(0, \eta_{1:T}, 0) \tag{119}$$

$$= C \sum_{j=0}^{\tau} \beta_j A^{t-j-1} \eta_1 \tag{120}$$

$$\leq C \sum_{j=0}^{\tau} \sum_{\ell=1}^{d} \beta_j \alpha_\ell^{t-j-1} v_\ell w_\ell^* \eta_1 \tag{121}$$

$$= C \sum_{j=0}^{\tau} \sum_{\ell=1}^{d} \beta_j \alpha_\ell^{t-\tau-1} (\alpha_\ell^{\tau-j} - \omega_\ell^{\tau-j}) v_\ell w_\ell^* \eta_1 \tag{122}$$

$$= C \sum_{j=1}^{\tau} \sum_{\ell=1}^{d} \beta_j \omega_\ell^{t-j} r_\ell^{t-\tau-1} (r_\ell^{\tau-j} - 1) v_\ell w_\ell^* \eta_1 \tag{123}$$

$$\|\hat{y}_t(0, \eta_t, 0) - y_t(0, \eta_{1:T}, 0)\|_2 \tag{124}$$

$$\leq \|\beta\|_\infty \sum_{j=1}^{\tau} \sum_{\ell=1}^{d} \frac{1 - r_\ell^{\tau-j}}{1 - r_\ell} \mu(r_\ell)_{t-\tau} \|C v_\ell w_\ell^* \eta_1\|_2 \tag{125}$$

$$\leq \|\beta\|_\infty R_\Theta \|\eta_1\|_2 \sum_{j=1}^{\tau} \sum_{\ell=1}^{d} \frac{1 - r^{\tau-j}}{1 - r} \mu(r_\ell)_{t-\tau} \|v_\ell w_\ell^*\|_2 \tag{126}$$

$$\|\hat{y}_{\tau+1:T}(0, \eta_{1:T}, 0) - y_{\tau+1:T}(0, \eta_{1:T}, 0)\|_2 \tag{127}$$

$$\leq \|\beta\|_\infty R_\Theta \|\eta_1\|_2 \sum_{j=1}^{\tau} \max_{r,\ell} \left\| \frac{1 - r_\ell^{\tau-j}}{1 - r_\ell} \mu(r_\ell) \right\|_2 R_\Psi \tag{128}$$

$$\leq \|\beta\|_\infty R_\Theta \|\eta_1\|_2 \left(\sum_{j=1}^{\tau} \sqrt{\tau-j}\right) R_\Psi \tag{129}$$

$$\leq \|\beta\|_\infty R_\Theta R_\Psi \tau^{\frac{3}{2}} \|\eta_1\|_2 \tag{130}$$

by the calculation in (52) and (53) because

$$\left\|\frac{1-r^k}{1-r}\mu(r)\right\|_2 \tag{131}$$

$$\leq (1-r^k)\sqrt{\frac{1}{1-r^2}} \leq \min\{1, k(1-r)\}\sqrt{\frac{1}{1-r}} \tag{132}$$

$$= \min\left\{\sqrt{\frac{1}{1-r}}, k\sqrt{1-r}\right\} \leq \sqrt{k}. \tag{133}$$

Combining (112), (118), and (130) using (106) and noting $\|\beta\|_1 \leq (\tau+1)\|\beta\|_\infty$ gives

$$\|\hat{y}_{1:T}(x_{1:T}, \eta_{1:T}, \xi_{1:T}) - y_t\|_2 \tag{134}$$

$$\leq \|\beta\|_2 \|\eta\|_2 + R_\Theta R_\Psi \sqrt{\tau}(\|\beta\|_1 + \|\beta\|_\infty \tau)\|\xi\|_2 + \varepsilon \tag{135}$$

$$\leq O(\|\beta\|_\infty \tau^{\frac{3}{2}} R_\Theta R_\Psi \sqrt{L}) + \varepsilon \tag{136}$$

$\square$

## C.2 Proof of Main Theorem

We prove Theorem 1 with the following more precise bounds.

**Theorem 15.** *Consider a LDS with noise satisfying the assumptions in Section 2.1 (given by* (2) *and* (3)*), where total noise is bounded by L. Then there is a choice of parameters such that Algorithm 1 learns a pseudo-LDS $\hat{\Theta}$ whose predictions $\hat{y}_t$ satisfy*

$$\sum_{t=1}^{T} \|\hat{y}_t - y_t\|^2 \leq \tilde{O}\left(Rd^{5/2}n\sqrt{T}\right) + O(R_\infty^2 \tau^3 R_\Theta^2 R_\Psi^2 L) \tag{137}$$

*where the $\tilde{O}(\cdot)$ only suppresses factors polylogarithmic in $n, m, d, R_\Theta, R_x, R_y, \log T$, and $R_1^3 R_x^2 R_\Theta^4 R_\Psi^2 R_y^2 \leq R$.*

*Proof.* Consider the fixed pseudo-LDS which has smallest total squared-norm error in hindsight for the noiseless LDS. Let $y_t^*(x_{1:T}, \eta_{1:T}, \xi_{1:T})$ denote its predictions under inputs $x_{1:T}$ and noise $\eta_{1:T}, \xi_{1:T}$ and $y_t(x_{1:T}, \eta_{1:T}, \xi_{1:T})$ denote the true outputs of the system. Given $\varepsilon > 0$, choosing $k, P$ as in Theorem 2 (Approximation) gives

$$\forall 1 < t < T, \quad \|y_t^*(x_{1:T}, 0, 0) - y_t(x_{1:T}, 0, 0)\|_2 \leq \varepsilon \tag{138}$$

$$\implies \|y_{1:T}^*(x_{1:T}, 0, 0) - y_{1:T}(x_{1:T}, 0, 0)\|_2 \leq \varepsilon\sqrt{T}. \tag{139}$$

By Lemma 14 (Perturbation),

$$\|y_{1:T}^*(x_{1:T}, \eta_{1:T}, \xi_{1:T}) - y_t(x_{1:T}, \eta_{1:T}, \xi_{1:T})\|_2 \tag{140}$$

$$\leq \varepsilon\sqrt{T} + O(R_\infty \tau^{\frac{3}{2}} R_\Theta R_\Psi \sqrt{L}) \tag{141}$$

In the below we consider the noisy LDS (under inputs $x_{1:T}$ and noise $\eta_{1:T}, \xi_{1:T}$). By Theorem 3 (Regret) and (141),

$$\|\hat{y}_{1:T} - y_{1:T}\|_2^2 \tag{142}$$

$$= (\|\hat{y}_{1:T} - y_{1:T}\|_2^2 - \|\hat{y}_{1:T}^* - \hat{y}_{1:T}\|_2^2) + \|\hat{y}_{1:T}^* - y_{1:T}\|_2^2 \tag{143}$$

$$\leq \tilde{O}\left(R_1^3 R_x^2 R_\Theta^4 R_\Psi^2 R_y^2 d^{5/2}n\log^7 T\sqrt{T}\right) \tag{144}$$

$$+ \varepsilon^2 T + O(R_\infty^2 \tau^3 R_\Theta^2 R_\Psi^2 L) \tag{145}$$

Choosing $\varepsilon = T^{-\frac{1}{4}}$ (and $k, P$ based on $\varepsilon$) means $\varepsilon^2 T$ is absorbed into the first term, and we obtain the bound in the theorem. $\square$

## Footnotes

[1]In other words, the minimal polynomial of $A$ divides $p$. For a diagonalizable matrix, the minimal polynomial is the characteristic polynomial except without repeated zeros.

[2]For a polynomial $p(x) = \sum_{j=0}^{\tau} \beta_j x^{\tau-j}$, let