[Reviews · NeurIPS 2018]

Reviewer 1



The submitted paper investigates spectral filtering for general linear dynamical systems (i.e., linear internal dynamics with disturbances and linear output feedback with measurement noise). Using convex programming, a polynomial-time algorithm for learning/predicting the behavior of latent-state linear dynamical systems (LDSs) whose state matrices are not symmetric is provided. Upper bounds on the learning/prediction performance are derived. Performance comparisons with the related algorithms are enclosed. The submitted paper is concise, well written, investigates an interesting topic, the proposed methodology is plausible, and the mathematics appears honest. Some minor comments are as follows: 1. Is the presented methodology applicable to continuous-time LDSs? Can (a class of) nonlinear dynamics be learned via the presented methodology? 2. At the end of the sentence in the 18th row, please provide a reference. The same applies to the sentence in the 22nd row. 3. The price paid for such strong theoretical results is found in the restrictive set of assumptions between row 92 and 102. In particular, assumptions 2, 3 and 5 appear to be rather restrictive. Is this correct? Is it possible to relax any of them? Is this a part of the future work as well? Please include these comments in the final version of the paper (if any).

Reviewer 2



This work develops a prediction method for the output of linear dynamical systems from past observations. The authors measure prediction accuracy in a regret setting. They introduced a class of pseudo-linear dynamical systems in which the output of the next observation depends on the previous tau observations and the entire history of inputs, and they showed that when such systems have poly(log T) many weights they can approximate linear dynamical systems (LDSs). The construction of the pseudo-LDS class relies on the spectral filtering technique introduced by [HSZ17], and the regret proof on the analysis of regularized follow the leader from online convex optimization. As far as I know the authors offer the first polynomial time algorithm for attaining sublinear-regret for predicting the output of linear dynamical systems without full state observation, i.e. with C not the identity matrix, and without a dependence on the spectral radius of the A matrix. Also, this work relaxes the requirement of A being symmetric, need in previous art [HSZ17], to A being diagonalizable; a difficult extension achieved by considering polynomials that have the phases of A's eigenvalues as roots. LDS represent a fundamental building block for continuous control problems, and there has been a lot of work recently in trying to develop data driven methods for controlling or identifying such systems from data. One difficulty in analyzing such methods is that the larger the spectral radius of A is the stronger the dependence on previous inputs and disturbances is. However, a larger spectral radius offers a larger SNR so it should not degrade performance. Algorithms and proof techniques that do not degrade as the spectral radius of A increases are valuable. My only complaint is the writing; it could be clearer, a weakness which decreased my score. Some line by line comments: - lines 32 - 37: You discuss how the regret cannot be sublinear, but proceed to prove that your method achieves T^{1/2} regret. Do you mean that the prediction error over the entire horizon T cannot be sublinear? - eq after line 145: typo --- i goes from 1 to n and since M,N are W x k x n x m, the index i should go in the third position. Based on the proof, the summation over u should go from tau to t, not from 1 to T. - line 159: typo -- "M" --> Theta_hat - line 188: use Theta_hat for consistency. - line 200: typo -- there should no Pi in the polynomial. - line 212: typo --- "beta^j" --> beta_j - line 219: the vector should be indexed - lines 227 - 231: the predictions in hindsight are denoted once by y_t^* and once by hat{y}_t^* - eq after line 255: in the last two terms hat{y}_t --> y_t Comments on the Appendix: - General comment about the Appendix: the references to Theorems and equations are broken. It is not clear if a reference points to the main text or to the appendix. - line 10: Consider a noiseless LDS... - line 19: typo -- N_i ---> P_i - equation (21): same comment about the summation over u as above. - line 41: what is P? - line 46: typo --- M_omega' ---> M_ell' - eq (31): typo -- no parenthesis before N_ell - line 56: the projection formula is broken - eq (56): why did you use Holder in that fashion? By assumption the Euclidean norm of x is bounded, so Cauchy Schwartz would avoid the extra T^{1/2}. ================== In line 40 of the appendix you defined R_x to be a bound on \|x\|_2 so there is no need for the inequality you used in the rebuttal. Maybe there is a typo in line 40, \|x\|_2 maybe should be \|x\|_\infty

Reviewer 3



Summary ------- This paper presents an algorithm for predicting the (one-step ahead) output of an unknown linear time-invariant dynamical system (LDS). They key benefits of the proposed algorithm are as follows: - the regret (difference between the prediction error of the algorithm, and the prediction error of best LDS) is bounded by sqrt(T) (ignoring logarithmic factors), where T is the length of the data sequence over which predictions are made. - the regret bound does not depend on the spectral radius of the transition matrix of the system (in the case that data is generated by a LDS). - the algorithm runtime is polynomial in the 'natural parameters' of the problem. The algorithm is based on a recently proposed 'spectral filtering' technique [HSZ17]. In previous work, this method was only applicable to systems with a symmetric transition matrix (i.e. real poles). The present paper extends the method to general LDSs (i.e. non-symmetric transition matrices, or equivalently, complex poles). The performance of the proposed algorithm is illustrated via numerical simulations, and is compared with popular existing methods including expectation maximization (EM) and subspace identification. Quality ------- In my view, the paper is of very high quality. The technical claims are well-substantiated by thorough proofs in the supplementary material, and the numerical experiments (though brief) are compelling and relevant (i.e. comparisons are made to methods that are routinely used in practice). Clarity ------- In my view, the paper is well-written and does a good job of balancing technical rigor with higher-level explanations of the key concepts. Overall, it is quite clear. I do have a few comments/queries: In my opinion, the supplementary material is quite crucial to the paper; not just in the obvious sense (the supplementary material contains the proofs of the technical results), but personally, I found some of the sections quite difficult to follow without first reading the supplementary material, e.g. Section 4.1. I very much like the idea of giving a high-level overview and providing intuition for a result/proof, but to be honest, I found it quite difficult to follow these high-level discussion without first reading through the proofs. This is probably the result of a combination of a few factors: i) the results are quite technical, ii) these ideas (spectral-filtering) are quite new/not yet well known, iii) my own limitations. This is not necessarily a problem, as the supplementary material is well written. Regarding the presentation of some of the technical details: Should \Theta be \hat{\Theta} in Line 188 and equation (5)? In the supplementary material, the reference to (equation?) (2) doesn't seem to make sense, e.g., line 33 but also elsewhere. Do you mean Definition 2? In Line 18, do you mean \sum_{j=0}^\tau\beta_jA^{i-j} = A^{i-\tau}p(A) = 0? In equation (10), is there an extra C? It's not a big deal, but i is used as both sqrt(-1) (e.g. (29)) and as an index, which can be a little confusing at first. Below (28), should M'_\omega be M'_l? Furthermore, should M'_l be complex, not real? If M'_l is real, then where does (31) come from? (I would understand if M'_l was complex). Are the r's in (38) and below missing an l subscript? Concerning reproducibility of numerical results: I would imagine that the numerical results are a little difficult to reproduce. For instance, the details on the EM algorithm (initialization method?) and SSID (number of lags used in the Hankel matrix?) are not given. Perhaps Line 174 'the parameter choices that give the best asymptotic theoretical guarantees are given in the appendix' could be a bit more specific (the appendices are quite large). Also, Unless I missed it, in Sec 5 k is not specified, nor is it clear why W = 100 is chosen. Originality ----------- The paper builds upon the previous work of [HSZ17] which considers the same one-step ahead prediction problem for LDS. The novelty is in extending this approach to a more general setting; specifically, handling the case of linear systems with asymmetric state transition matrices. In my view, this represents a sufficiently novel/original contribution. Significance ------------ The problem of time-series prediction is highly relevant to a number of fields and application areas, and the paper does a good job of motivating this in Section 1. Minor corrections & comments ---------------------------- Full stop after the equation between lines 32-33. Line 39, 'assumed to be [proportional to] a small constant'? It's a little strange to put Algorithm 1 and Theorem 1 before Definition 2, given that Definition 2 is necessary to understand the algorithm (and to some extent the theorem). Line 185 'Consider a noiseless LDS'.